# Cortical Visual Impairment in Childhood: ‘Blindsight’ and the Sprague Effect Revisited

**DOI:** 10.3390/brainsci11101279

**Published:** 2021-09-27

**Authors:** Gerry Leisman, Calixto Machado, Robert Melillo

**Affiliations:** 1Movement and Cognition Laboratory, Department of Physical Therapy, University of Haifa, Haifa 3498838, Israel; drrm1019@aol.com; 2Department of Neurology, University of the Medical Sciences of Havana, Havana 10400, Cuba; 3Institute of Neurology and Neurosurgery, Havana 10400, Cuba; braind@infomed.sld.cu

**Keywords:** cortical blindness, cortical visual impairment, vision, blindsight, Sprague Effect, infant vision

## Abstract

The paper discusses and provides support for diverse processes of brain plasticity in visual function after damage in infancy and childhood in comparison with injury that occurs in the adult brain. We provide support and description of neuroplastic mechanisms in childhood that do not seemingly exist in the same way in the adult brain. Examples include the ability to foster the development of thalamocortical connectivities that can circumvent the lesion and reach their cortical destination in the occipital cortex as the developing brain is more efficient in building new connections. Supporting this claim is the fact that in those with central visual field defects we can note that the extrastriatal visual connectivities are greater when a lesion occurs earlier in life as opposed to in the neurologically mature adult. The result is a significantly more optimized system of visual and spatial exploration within the ‘blind’ field of view. The discussion is provided within the context of “blindsight” and the “Sprague Effect”.

## 1. Introduction

We intend to examine the differences between cortical visual impairment (CVI) in early life, compared to adult-onset CVI in the context of our current understanding of age-related neuroplasticity of the visual system. We intend to examine why it is that recovery of vision after visual system insult early in life has a greater chance for recovery as compared to insult occurring later in neurological development or maturity.

A principal reason for visual dysfunction in childhood in developed countries is CVI [1,2,3]. This has occurred as technology development has led to better visual treatment for other conditions such as congenital glaucoma, retinopathy of prematurity, and congenital cataracts as well as the increased survival of infants with central nervous system damage or disease.

The incidence of CVI has increased, with it now being a highly significant public health concern. Approximately 30–40% of children with visual impairments have CVI. The National Institutes of Health website cites a CVI prevalence of 10.5% of all children with developmental disabilities [4]. Generally, the prevalence of visual impairment in children under 16 years ranges between 10–22 per 10,000 births in developed countries and 40 per 10,000 births in developing countries [5,6].

In children with cerebral palsy, approximately two-thirds also demonstrate impaired visual acuity and/or field defects indicative of CVI [6]. In an African study, 47.7 percent of cerebral palsied children also demonstrated CVI [6,7], and in India, reports of 28 percent have been described [6,8].

An infant or child is said to’ have CVI if (a) the loss of functional vision cannot be explained completely by an eye examination; (b) has a history of neurological dysfunction even with brain imaging studies appear to be normal; (c) demonstrates an array of visual or behavioral features identified in medical, psychological, or educational research [9]. As CVI is a consequence of brain insult rather than ocular dysfunction, an understanding of the dynamic properties of neurological development of the infant and child can assist in planning and developing better treatment protocols that may influence the developing child’s functional vision. The process of neuroplasticity related to the development and function of the visual system will be discussed.

Damage, insult, or dysfunction to the visual system during fetal, neonatal and infant development may well have long-term consequences that are, as we shall see, potentially more capable of alteration and restoration of function in the infant and child as compared to similar insult in adults [10].

## 2. Neuroplastic Changes in the Visual System

### 2.1. The Process of Neuroplasticity

We have developed a better understanding of what is termed neuroplasticity or the ability of the brain or reorganize itself after responding to a range of sensory experiences. We have had a long-standing notion that the changeable or plastic nature of the brain is limited to critical periods in early child development [11,12]. However, we have recently come to realize that numerous regulators are capable of modulating neuroplasticity beyond childhood, and which are variable from one individual to another. Among the factors that can contribute to variability in the function of neuroplasticity include but are not limited to inhibitory function of brain networks, age, sex, type of brain insult or damage, psychological factors, and neuromodulatory systems and function. With a better and more comprehensive understanding of the nature of experience-dependent neuroplasticity, we will be better able to exploit and utilize the function for more effective learning and recovery of functional abilities after brain insult.

Of singular importance to the effective employment of neuroplasticity in early life are critical periods for neuronal connectivities. Studies of critical periods have indicated the significance of sensory experience for effective neurological development and sensory function. The nature of sensory experience can stimulate numerous effects on critical periods’ outcome and duration. By enriching sensory environments, one may be able to extend the plasticity associated with critical periods [13] or kindle dendritic growth [14,15], and enhance neuronal responsivity [16,17] whereas damaged, deprived or unstructured environments or stimulation can defer the onset of critical periods and create an environment in which cortical neurons remain in an immature state [18,19,20]. We learn from these studies that there exists a critical role of neuromodulator systems as the main gating mechanisms of plasticity, as well as maintaining a significant function in shaping cortical connectivities and cognitive abilities.

One aspect of neuroplasticity that has been reported to affect visual function developmentally is perceptual learning whereby the repeated practice of a visual task is associated with significant and enduring improvements in visual performance [21]. Visual perceptual experiences can affect plastic modifications in the optic pathways and visual system as demonstrated by observable effects on V1 activation in adults observed with *f*MRI [22,23,24,25,26,27].

Retinotopic blood oxygenation level-dependent signal (B.O.L.D.) response increases have been noted after learning. This finding offers support that learning supports visual cortical activity with tasks that allow the distinction of trained targets from background noise [28]. The change has been observed both in adults and in children and is found exclusively in the trained eye after numerous iterations [29]. Training has also been observed to affect Vernier acuity (small differences perceived between two lines), orientational selectivity, feature detection, and differences in depth [30]. It appears then that visual perceptual learning can be facilitated by response strength characteristics or individual neuronal tuning rather than by large-scale cortical network reorganization [29].

The beginning of visual system plasticity’s critical period commences with experience-dependent GABA-mediated inhibition in the fetal environment [31]. In transgenic animals with an absence of an isoform of GABA, monocular deprivation change in the responsivity of the visual cortex was evidenced [32]. Inhibitory transmission reduction during the early stages of development halts the onset of the critical period for visual cortical neuroplasticity [31]. The limited plasticity in the adult visual cortex can be enhanced by previous visual deprivation, which is associated with a loss of GABA receptors, and reduced by GABAergic modulators [33]. It has been shown that a brief reduction of GABAergic inhibition in the brains of rats can reopen a window of plasticity in the visual system a long time after the normal closure of the critical periods [22].

Visual cortical circuitries are remodeled by visual sensory experiences and are maintained throughout one’s life by extracellular molecular factors that restrict the nervous system’s processes of neuroplasticity [34]. The adrenergic system, for example, impacts neuroplasticity significantly [35]. This system can dynamically play with reducing or prolonging plasticity in the nervous system. Serotonin reuptake inhibitors can prolong plasticity and calcium channel blockers can diminish neuroplasticity [35]. With dysfunction cholinergic input in rat visual cortex, previously learned pattern discrimination is not affected but fine discrimination is in rat visual cortex allowing the conclusion that acetylcholine (ACh) can facilitate neuroplasticity and cholinergic function impairs the process [36,37]. In functional vision, ACh supports neuroplasticity in V1 and participates in spatial mapping functions in other cortical regions [37]. Such mediators regulate complex central nervous system functions that include brain plasticity, cognitive processes, and behavior [38]. Another important mechanism concerns cAMP Response Element-Binding Proteins (C.R.E.B). This transcription factor activity is stimulated with monocular deprivation in juveniles and wanes with visual cortical maturation [39].

On the other hand, structural plasticity in the human primary visual cortex is not without controversy. Both the time course and location of structural changes differ significantly between reports, despite the correspondence between training paradigms [40]. Additionally, voxel-based morphometry employed as a means of studying structural changes in the brain is questionable [40,41]. Studies involving cortical plasticity in humans with retinal lesions do have important limitations, as it is difficult to exclude changing borders or spared retinal regions in the absence of histological examination [41,42]. V1 responses in the presence of central retinal lesions can also be the result of activation of the extrastriate cortex or subcortical structures [41,43]. Even with the absence of significant structural remodeling in adult life, the reorganization of cortical connections in terms of loss of dendritic spines and growth may be the structural basis for experience-dependent plasticity [38,39].

There exist numerous examples of pathological neuroplasticity that include but are not limited to pathological pain [44], dementia [45], and schizophrenia [46].

Unfortunately, there is a paucity of studies of pathological neuroplastic effects in the developing brain. The literature on significant prenatal stress is another instance in which neuroplasticity has been demonstrated to affect prefrontal area neuronal complexity [40] with negative effects on cognitive and motor function both in brain development as well as in adults [41]. While it is not clear how these negative mechanisms work developmentally, we know, at least from the postnatal stress literature that stressors have an effect of altering gene expression in the brain [42,43] and there have been reports of brain volume loss and modified white matter after brain trauma. In adults, the effect has been to “age the brain” inducing neurodegeneration associated with poor functional and cognitive outcomes [47] but we do not know how negative neuroplasticity plays out in infancy.

Therefore, the neurophysiology of visual plasticity involves numerous processes which are not completely understood and are therefore controversial Understanding these mechanisms will be important for better recognition of the occurrence of neuroplasticity and for developing treatment paradigms. Wandell and Smirnakis [41], had indicated that “It is not worth having a debate as to whether the brain is plastic or not: it is both. It is more important to study the conditions under which each system is stable or plastic.”

### 2.2. Neuroplastic Changes in the Visual System after Early Insult

During the third trimester of pregnancy, the vulnerability of white matter damage can be significant as the optic radiations may be more sensitive to insult. This is not infrequently evidenced by unilateral periventricular hemorrhagic infarcts as well as by bilateral ischemic lesions [48,49] It is during the third trimester of pregnancy that we note the significant degree of plasticity in the development of thalamocortical afferents, as it is that during this period, with relevant axonal guidance, there still exists migration of afferents from the sub- to the cortical plate [50].

Seghier and colleagues [51,52,53], studied a three-month-old infant who had sustained a perinatal left arterial stroke with damage to the optic radiations but with sparing of the primary visual cortex (PVC) or V1. Initial testing revealed cortical activation on the non-lesioned side and no evidence of the optic radiations on the ipsilesional side. When a follow-up study was performed at twenty months of age, *f*MRI demonstrated clear activation on the ipsilesional side representing neuroplastic changes in the thalamocortical pathway with attendant functional connectivities in the lateral geniculate [54].

Guzzetta and colleagues [55] found support for the notion that infants with preterm unilateral lesions in periventricular white matter regions that occurred early in the third trimester of pregnancy, evidenced normal visual fields independent of the fact that a significant lesion was evidenced in the left periventricular region where the optic radiations should have been found. Their findings in their case are reproduced in Figure 1.

Numerous investigators [57,58,59,60,61,62,63,64,65,66] have supported the notion that preterm infants having sustained lesions in periventricular white matter evidenced somatosensory projections that found alternate routes that provided linkages with the postcentral gyrus. Guzzetta et al. [56] reported on a patient demonstrating normal visual fields despite the existence of a lesion of the left periventricular white matter involving most of the tissue where optic radiations would normally sit (as described in Figure 1). These investigators demonstrated, through diffusion tensor tractography, how optic radiations possessed projections that deviated away from the affected area and instead bypassed a cystic lesion eventually connecting with the occipital cortex. In addition to the *N* = 1 report of Guzzetta and colleagues [55], post-mortem support for the notion of rerouting during the last trimester of pregnancy is supported by Hevner [59] and others in animal studies [60,61,62,63,64,65].

It appears then that early insult to the PVC or optic pathways provides the infant with a critical period of developing neuroplastic changes that are better capable of allowing for the development of functional vision and with support for the conclusion that the child’s brain has a better capacity for rewiring than that of the older brain [66,67,68].

## 3. Adult vs. Child Cortical Visual Impairment

We here examine the significant differences between child and adult understanding of neuroplasticity as it relates to early vs. later life CVI.

We have learned a lot in recent years about visual system neuroplasticity, which has led in part to the replacement of the term “cortical visual impairment” with “cortical blindness” [68]. It is not an idle distinction but important in that it made contradistinction between the blindness observed in adults who had sustained traumatic injury or stroke involving the visual system as opposed to children having sustained the same or similar damage congenitally.

Now that we have gained the ability to examine the visual system in neonates and infants and of late in the fetus, with tools such as 4-D ultrasound, contrast sensitivity using eye movements, *f*MRI, magnetoencephalography (MEG), and the like [69], we can now better understand the nature of network communication in the brain allowing more in-depth study of developmental network dysfunctions [70,71,72].

While most neonates are born healthy and well, surprisingly few have developmental trauma or disabilities. However, there exist numerous possibilities in which the neonate can be exposed to potentially damaging circumstances resulting in brain dysfunction or damage. A great deal has to do with the time at which the infant is exposed to harmful events [73,74]. If damage occurs during the first trimester of pregnancy cortical development is likely to be significantly affected [75,76]. The anatomy of the damage does not necessarily have a relation to the associated functional visual dysfunction for numerous reasons that we shall explore further later. One of the reasons, however, is that non-visual tissue can subserve aspects of visual functioning, which can in turn allow for the sparing of functional vision [77].

During the third trimester of pregnancy, an infection could influence brain-based visual function in turn associated with ventricular white matter abnormalities which then could affect the geniculostriatal system which we will examine further when discussing the Sprague Effect [78,79].

There is the possibility that, because of genetic or ischemic incidents, preterm neonates may suffer from stroke [80]. In this population, inconsistencies are often found between behavioral testing of visual function and magnetic resonance imaging (MRI) results [80,81,82]. Surprisingly, directional eye movements and visual acuity can remain intact while dysfunction of other aspects of function vision may be evidenced, such as disorders of peripheral vision. Mercuri and colleagues [83] noted significant improvement in CVI neonates when retested during the school years. Crawford and Golomb [84] noted, in a structured literature review, that visual field defects were the most frequently noted visual deficits after perinatal and childhood stroke. Conclusions are difficult to draw as there exists a significant lack of literature and the literature that does exist largely consists of individual case reports. Children can experience significant visual morbidity after stroke but can also have a significant potential for recovery.

The hypothesized mechanism for the relatively low prevalence of visual abnormalities post-stroke in children as compared to adults is assumed to be due to neuroplasticity within the visual system [76,85]. This is supported by evidence of intact visual function in children with frank damage to the visual pathways and cortex [44,86,87]. Congenital or neonatal brain damage invariably leads to bilateral injury commencing in the periolandic areas and involving gray matter and cortical regions of the brain [71,88,89].

Unlike adults who have sustained similar types of neurological insult, we make a case that neonates and infants have a greater likelihood of recovery of functional vision as the developing brain relies more on the effective configuration and functioning of brain networks in contradistinction to adults for whom circumscribed visual system lesions are more likely to have a more permanent effect [88,90,91,92]. Additionally, the area of the brain that contributes significantly to visual dysfunction in childhood involves the thalamus and basal ganglia which is evidenced even with the absence of significant cortical area damage or dysfunction [84,93]. This is not the case in adults. Overall, we can state that based on the non-extensive literature on the subject of the effects of early brain damage on vision, we can indicate that neonatal visual dysfunction is not uncommon. However, the relationship between the location of a lesion in the visual system of neonates and infants is not as clearly demarcated as it is with individuals sustaining such damage later in development and certainly in adulthood [94,95,96,97]. The indications are therefore that the child has a greater chance of recovery of functional vision after damage to the visual system than does the adult.

## 4. Is Recovery of Normal Conscious Vision Possible?

Our visual perceptual abilities are dependent on the pathways represented in Figure 2 [97,98,99]. Guzzetta and associates [43,94] propose that three criteria are necessary for the restoration of vision that includes: (a) pathology of involvement of the geniculostriatal pathway, (b) specific loss of vision that is independent of any other functional abnormality, and (c) regaining the formerly impaired function with concomitant empirical change in brain state or electrophysiological activity.

As indicated in Section 3, adult functional vision is significantly different than that of the child with, among other things, evidence that the type of neuroplastic reorganization allowing for visual perception in the blind field seen in childhood cortical visual area insult has no adult analog [100]. A restricted broadening of the visual field can be normally seen shortly after the cerebral insult. This is a consequence of the resolution of temporary dysfunction of perilesional regions [84,101] or from modifications like the neural pathways adjacent to the lesion, including the size of the receptive field [84]. Beyond spontaneous recovery, partial or otherwise, support exists that interventions exist that can positively affect the size of the visual field through the recruitment of additional less efficient yet still intact visual pathways [84,102,103,104,105].

On the other hand, Bouwmeester and colleagues [106] found no support that limited broadening of visual field results in improved ocular motor scanning strategies or improvement in activities of daily living. This may be a consequence of the paucity of studies examining functions such as contrast sensitivity or line and edge detection in stroke in childhood or the lack of data on blindsight and certainly on the Sprague Effect in childhood CVI. The functional vision adaptation and visual pathway alterations noted after cerebral insult are likely not the result of direct recovery of lost vision, but rather the effect of learned compensatory visual-motor strategies [84,107].

### Neuroplasticity and Developmental Damage to the Primary Visual Cortex (V1)

Our essential question is whether there is evidence to support the reestablishment of visual function when the insult occurs in infancy or early childhood? Related to developmental insult from animal studies, we have learned that lesions in early development generate greater recovery of functional visual capacities when compared with lesions acquired later in the life cycle.

The functions studied in cats and monkeys have included significantly enhanced performance in the discrimination of motion [108], shape [109], depth [110], and visual orientation [111], with early lesions in both animal and human revealing significantly greater neuronal rewiring and reorganization with lesions early in development [104,112]. Teuber, already in the 1970s [113] had examined individuals with lesions in the occipital regions resulting from an injury that had occurred between adolescence through the twenties. Teuber found a relationship between the age at which the insult had occurred and the degree of shrinkage of the scotoma.

We know that children who have undergone hemispherectomy for conditions such as Rasmussen’s syndrome [114] in early childhood fare better and, considering the drastic surgery, have relatively little functional impairment when compared with adults having undergone similar procedures [115,116]. There is much support since then for the notion that early developmental lesions, especially those that are congenital, are significantly more disposed to neuroplastic restructuring with the developing nervous system being more likely to employ compensatory pathways to counteract the effects of damaged brain areas thereby restoring functional vision.

Of great relevance to our argument of functional neuroplastic changes in early visual system insult, are the numerous reports of neonatal visual system damage highly associated with compensatory functions that are normally controlled by damaged regions of the brain [117,118]. A case was reported by Werth [119] of a hemispherectomized infant of 4 months who had later demonstrated intact visual fields. One reason that might support the neuroplastic changes in the pathways involved in hemianopsia in infancy maybe that area V1 of the integrated intact hemisphere also acts in response to the simulation of the ipsilateral blind hemifield. Cornwell and associates [120,121] proposed that a separate reorganizational pattern is evidenced in congenitally acquired lesioned individuals as opposed to lesions acquired later in life. Possible neuronal rearrangement patterns are represented in Figure 3.

Supporting the above arguments are animal studies in which cats lesioned in infancy demonstrate no significant difference with intact cats for complex pattern, shape, and hidden figure discrimination in contradistinction to adult-lesioned cats [120,121]. These findings have been confirmed by others [122,123]. Moore and colleagues [124] also noted that early lesioned monkeys demonstrated normal detection ability shortly after the lesion and recovered function relatively quickly thereafter. In subsequent studies, Moore and colleagues [125] also noted that monkeys with early striatal lesions demonstrated oculomotor movements appropriate to the direction of the stimulus presentation having seemingly had motion detection within the scotoma spared.

Also, in the 1990s, Mercuri, and associates [126] found significant visual field deficits with kinetic in infants who had suffered arterial stroke perinatally. When these investigators performed follow-up examinations of the same infants when they reached school age, no evidence of visual field defect was noted [84,127]. There are numerous possible explanations for the result, including geniculostriatal pathway changes, which will be discussed more fully later in the context of the Sprague Effect. However, as it is quite possible that infant visual field testing is performed by directing focus to the midline, the infant must uncouple attentiveness from the central field to a novel peripheral stimulus thus allowing the non-damaged visual cortical pathways to support a shift in attention. The findings reported in the follow-up studies of Mercuri and associates [83] may have been the result of the development or the capacity to modify the focus of attention as opposed to the expansion of the visual fields.

Reinforcing this notion is the fact that of the children studied, all of whom demonstrated deviant visual filed during their first year, also presented with parietal lesions evidenced by MRI with sparing of the optic radiations and the primary visual cortices [83]. We can add that there are multitudes of mechanisms of visual development occurring at different developmental stages confounding comparison between components of recovery.

A singularly important question is that when there exists physical damage to the visual pathways in infancy or early childhood, is it possible to know whether the maintenance of normal vision is a function of neuroplastic reorganization or rather a result of some other process? Developmentally we know that occipital lobe structural abnormalities do not necessarily universally produce defects of the visual fields [127,128,129,130] as it is known that with developmental cortical deformities, neuroplastic mechanisms are likely to correct such problems early in development.

We can presume that the mechanism subserving the reorganization of pattern detection relates to the effectiveness of the tissue surrounding the visual system lesion as represented in Figure 2. Dumoulin and associates [76] had found involvement of dysplastic tissue active in visual information processing assumed to be the result of neuroplastic network reorganization. These results have been observed by numerous other authors in both adults [131,132,133,134,135,136,137] and especially in children [117,138,139]. The consistently observed degree of functional residual vision found in early lesioned animals is related to a substantial visual system rewiring and of neural system activity adjustment.

## 5. Blindsight

When one compares adult-onset visual system-lesioned individuals with those whose lesions having originated in childhood, there appears to be a significantly greater subjective cognizance of stimuli in the blind field. Damage to the geniculostriatal pathway and the PVC (V1, striate cortex) in adults will result in CVI. Studies by Larry Weiskrantz [140,141] demonstrated that some occipital lobe damaged patients showed a degree of visual responsivity and visually guided behavior in a blind field to a degree greater than that which could be explained based on chance.

Although the PVC (V1) is a requisite fundamental staging area for visual information processing, individuals with lesions of V1 often have substantially spared visual function [140,141,142,143,144,145]. Consistent with a key role of V1 in visual awareness, residual vision for these individuals is associated with a lack of consciousness, a condition termed blindsight [139]. Subjects with blindsight can direct their eyes toward visual stimuli presented within the scotoma [142].

We know then that blind individuals possess vision, blind fields as well as the perception of visual fields with shades of gray. Blindsight individuals are capable of appropriately responding to stimuli without being aware of those stimuli. When these individuals respond to visual signals, they report seeing nothing. This may be described as unconscious vision or vision without knowing. The regions of the visual field with residual vision are those where individuals inconsistently respond, possess a significant degree of variability, demonstrate significantly increased thresholds, and increased reaction time as reflected in Figure 4.

In an attempt to explain the nature of blindsight, visual perception in the blind field may be aided through a functioning white-matter pathway that exists between the V5/hMT+ region and the lateral geniculate nucleus. It should be noted in this regard that V5/hMT+ reportedly has a causal role in tactile direction processing and strengthens the case for V5/hMT+ serving multimodal motion perception, which may influence the phenomenon of blindsight.

We know that congenital blindness in early childhood in the V1 region can reverse the loss of functional vision and sustain visual perception [44]. Congenital, childhood, or adult V1 insult implies the contribution of the striatal areas and dorsal stream that relates to the control of visually guided behavior. While the effects of damage to the V1 area is now well represented in the literature [145], there exists ample clinical data to support the notion that restoration of function resulting from neuroplastic changes are the result of activation of secondary visual pathways [44,145,147,148,149,150]. The presumed pathways that enable blindsight may be viewed in Figure 5.

### 5.1. Differences between Early Life Visual System Insult and Adult-Onset Blindsight

#### 5.1.1. V1 Injuries in Childhood in the Context of Blindsight

We have learned that both primates and humans with acquired V1 damage congenitally or in early child development often demonstrate significantly greater degrees of residual vision [119,151,152] indicating the importance of the developing brain in the acquisition of blindsight. Mundinano and colleagues [153] reported on a two-week-old neonate with significant V1 damage. Functionally, the infant demonstrated significant conscious visual function with augmentation of the inferior pulvinar-mid-temporal pathway. It seems, therefore, that the pulvinar has a significant influence on generating blindsight with a potential contribution of the geniculostriatal pathways [122,154]. Recurring visual stimulation within V1’s blind field can resuscitate remaining functional vision [155] possibly through the geniculostriatal system. While the geniculostriatal pathways are principally involved in blindsight in congenital, neonatal, or infant trauma to the visual system, mixed results have been found in adult-onset injury in which functional connections to the lateral-geniculo-MT pathways in the damaged hemisphere singularly govern the manifestation of blindsight [156]. This indicates that pulvinar and lateral-geniculate connections to the MT region are a function of the stage of life when V1 lesions occur.

In contradistinction to the above conclusion, Yu and colleagues [157] found that lateral-geniculate pathways connected to the extra-striatal cortical regions could be effective in preserving function in V1 independent of age. Yu et al. [157] concluded that even with an insult to the lateral geniculate, the remaining intact neurons might have sufficiently normal properties to allow for blindsight.

#### 5.1.2. V1 Injuries in Adult Brains and Blindsight

We have already determined that the brains of infants and children possess a significantly greater degree of plasticity and therefore possess a greater degree of likelihood of functional recovery when compared with adults who have sustained similar types of insult to the visual system.

When comparing adults in this way, unilateral V1 insult has been hypothesized to result in the non-involved hemisphere’s capacity to support blindsight. On the other hand, there are ample reports that do not support such a hypothesis as discrete and parallel pathways support neuroplasticity and restoration of functional vision in adult populations [152,156]. There is support for the notion that with a unilaterally intact contralateral V1, there is a likelihood for extrastriatal stimulation or PI, LGN, or SC stimulation. This notion has been suggested by Ro and Rafal [147] by employing transcranial magnetic stimulation (TMS) to create temporary and reversible lesions. There has also been much reported on the effects of TMS on conscious and unconscious visual perception of moving objects resulting from TMS of the V1 regions resulting from extrastriatal areas [158,159,160].

These findings suggest that the mechanisms of blindsight in patients who suffered V1 lesions as adults might be different from those supporting unconscious motion processing in neurologically intact individuals. While there is a paucity of controlled studies on the subject, there do exist ample clinical findings to indicate the existence of alternative visual pathways supporting functional visual plasticity in childhood. Guzzetta and associates [43,47] and Boyle et al. [161] have noted and discussed the distinctions of children who can perform visual discrimination tasks by rerouting processing through the ventral stream. These investigators have provided flash visual evoked response data supporting increased PI-MT processing with blindsight in childhood-onset insult compared to adult-onset damage (also see Fox et al. [149]).

#### 5.1.3. Mechanisms of Difference in Children vs. Adult Mechanisms in Blindsight

The literature on blindsight has distinguished between the effect with (Type II) or without (Type I) cognizance of the stimulus [162,163]. However, forced-choice testing results are comparable for the different forms of blindsight [163]. While there is agreement that the phenomenon of blindsight exists, its neural underpinnings are still not adequately understood and especially not in childhood, and its relation to neuroplasticity itself is not adequately agreed upon among investigators [77,164].

Bridge and colleagues [105] have, in a case study, provided some evidence that indicates the likelihood of significant functional reorganization of visual pathways in a 53-year-old adult male hemianopic individual with a right-sided V1 lesion. The patient showed atypical contralateral connections linking the right lateral geniculate nucleus and the left MTþ/V5, in addition to connectivities between the two MTþ/V5 areas found to be absent in control participants, with the aberrant connectivities circumventing the calcarine cortex. Additionally, Leh and colleagues [113] found atypical connectivities from the superior colliculus that projected to primary visual areas as well as parietal associative areas.

In attempting to understand the clinical literature of adult vs. child blindsight and vision recovery, the literature supports the notion that the superior colliculus and lateral geniculate are significantly involved in neuroplasticity associated with blindsight [111,162,163] supporting earlier evidence from monkey and cat lesion studies [165,166,167,168,169]. As there is evidence to support the rewiring of visual cortical circuitries in adults with visual system lesions, visual exercises have been reported to exert an effect on adult V1 lesions with training employing ipsilateral visual field stimuli with the effect being a restoration of visual consciousness [154]. However, infants and children have a significantly greater likelihood of more comprehensive blindsight and even the restoration of visual function which is likely associated with the exuberant connectivities found in early childhood brain development [2,4]. Finally, Bridge and associates [105] have offered compelling evidence, based on MRI tractography, that after visual pathway damage in childhood, alternative routes for visual information processing may either establish or strengthen connectivities. She and her colleagues demonstrated that their patient, unlike controls, demonstrated a contralateral pathway from the right LGN to the left MT+/V5 and additionally a significant cortico-cortico connection existed bilaterally between MT+/V5.

## 6. Sprague Effect

Besides understanding developmental aspects of visual neuroplasticity, we are also examining the Sprague Effect [78] in which small tectal lesions can reestablish visual orientation in the half field contralateral to the lesion in animals with significant unilateral geniculostriatal impairment. We provide support by illustrating the case of an adult with a midbrain tumor who had developed a left-field hemineglect, which was subsequently corrected after evidence of damage to the superior colliculus on the contralateral side to the lesion. This case demonstrates a human analog to the Sprague Effect that has been reported in non-human studies. The explanation of the effect in humans and, which can be applied developmentally, is that a reduction in right tecto-pulvinar activity (consequent to the additional right collicular damage) can provide compensation by increases in thalamic reticular nucleus (TRN) suppression of left pulvinar activity.

The thinking, ultimately supported by the Sprague Effect, was originally proposed by Denny-Brown’s [170] primate studies in which he found that superior colliculus injury is highly associated with contralateral hemispatial neglect. Later, Sprague [78] produced a right posterior cortical lesion and ablated the left superior colliculus in cats thus producing right hemispatial neglect.

More specifically, the Sprague Effect can be better understood by an understanding of spatial neglect, a failure to react to novel stimuli presented to aspects of visual space and without being able to attribute the lack of responsivity to a specific lesion [171]. The source of contralesional hemispatial neglect has been attributed to posterior temporoparietal defects. The point of connection between contralesional hemispatial neglect and the Sprague Effect is that Sprague noted that with the excision of a cat’s right posterior cortex, post-surgically, the cats demonstrated a defect in visually guided behavior in the contralesional field of view in a similar fashion to what is seen with hemispatial neglect. Sprague had noted that the cat exhibited improvement in visually guided behavior after additional damage in the form of excision of the superior colliculus contralateral to the lesion. Sprague’s hypothesis about the source of the defect was that the neglect was a function of the dysfunction of the colliculus ipsilateral to the induced lesion, thereby creating significantly increased inhibition from the contralateral colliculus. Therefore, it can be hypothesized that elimination of the contralateral tectum reduced inhibition and permitted the functioning of the colliculus on the ipsilateral side of the lesion.

Animal models and human anatomical preparations [172,173] have demonstrated that there exists a significant aggregation of decussating retino-colliclar fibers from each retina that project to the contralateral superior colliculus when compared to fibers that project to the ipsilateral superior colliculus. When this is considered in the context of the Sprague Effect, monocular occlusion, as an interventional strategy, may decrease activity in the collateral tectum and thereby reduce activity in the retino-colliclar pathway, supporting functional neuroplasticity and offering a therapeutic effect [174]. Therefore, considering the Sprague Effect, the eye ipsilateral to the lesion could be patched, theoretically diminishing activity level in the contralateral superior colliculus. As each of the superior colliculi can trigger the ipsilateral hemisphere, patching the ipsilesional eye would have the effect of diminishing the activity level of the contralesional superior colliculus theoretically resulting in the decrease of hemispheric activation asymmetry.

While there have not been many cases reported of such an effect, this form of neuroplasticity has been found in a 12-year-old female who demonstrated a right-sided cortical dysplasia consequent to seizure activity [175]. Perimetric testing revealed normal visual fields however, visual evoked responses indicated that the P100 response in leads over the lesioned hemisphere were displaced towards adjacent temporal and parietal areas. *f*MRI additionally demonstrated cerebral activity in the same cortical areas thereby intimating a displacement of visual regions peripheral to the striatal region.

Lambert and colleagues [176] had studied a 9-month-old infant where MRI indicated the existence of right hypoplasia of the occipital cortex but with no discernible visual defect or dysfunction as measured by flash visual evoked potentials. The lack of further analysis prevented a more universal understanding of the nature of plastic changes. The overlying conclusion, however, both in the case of Lambert and associates as well as from others is that functional vision, at least in infancy and childhood can be supported by pathways outside of the PVC even though we are still scratching at finding the mechanisms by which that happens.

As there exists clear support for residual unconscious processing of position, orientation, and motion of visual stimuli displayed in the scotomas of congenitally lesioned children that are not found in the children with acquired lesions, the literature suggests that congenital lesions can trigger significant reorganization of the visual system to improve function after early brain insult. Recognition, detection, and use of ‘blindsight’ and the possibility of the existence of the Sprague Effect in children may aid us in developing strategies to facilitate interaction with these children’s surroundings. Such mechanisms may well assist in the development of interventional strategies capitalizing on neuroplastic network development during critical periods of infant and child development.

The notion that early PVC insult can occasion transposition of function to alternate brain areas, which normally would not be responsible for visual processing is intriguing. Amir Amedi and colleagues have found much support for the auditory presentation of letters, words, and pictures being able to activate visual areas of the brain [177,178,179].

Further, Pascual-Leone and Amir Amedi [180] had studied Eşref Armağan, a blind Turkish painter born with *anophthalmia* (the absence of eyes) and, therefore without sight. They were interested in examining neuroplastic changes in the connections within Armağan’s visual system. The congenitally blind painter taught himself to paint natural scenes and understood perspective. Additionally, and regarding Eşref Armağan, John Kennedy [181] indicated that the congenitally blind can develop functional visual concepts without visual experiences and those behavioral outputs are reflected in activity in the occipital areas of the brain. All of those who have studied this case agree that with blind persons, the visual system functions differently when compared with sighted individuals. When Amedi and colleagues further analyzed the results, they found that Armağan’s visual cortex was activated during the drawing task, but hardly at all for verbal memory, indicating rerouting of networks. In scans that were held while Armağan drew, his visual cortex was activated [177,182,183,184].

Pascual-Leone and colleagues [180] had reported that when studying scribbling and drawing in the cortically blind, the PVC was activated with drawing tasks and concomitantly, brain regions involved in line and edge detection. The implication from these findings is that the visual cortex itself may be a type of “spare computer” with a bias for both haptic spatial as well as visual processing tasks. The same brain regions are involved in drawing for both sighted and blind children although the pathways within the system may be different.

## 7. Possible Interventions in Childhood CVI

McDowell [185] and Chang and Borchert [1] have indicated that CVI is the most common cause of visual impairment affecting children in both the developing and developed world. The clinical literature offers little by way of intervention strategies for infants and young children who have sustained CVI. However, of recent interest is the study by Ben Itzkak and associates [186] in which Clinical records of 630 children of median age 77 months with suspected for CVI were reviewed. Their Delphi study indicated the existence of seven visuoperceptual dimensions, that discriminated between the presence or absence CVI. The dimensions included: object-picture recognition, visual-spatial perception, visual discrimination and matching, and figure-ground perception. The study indicated that visual-perceptual schema could characterize dysfunctional and intact functions. These authors indicated that visuoperceptual profiling can serve as a basis for individualized interventions in heterogeneous disorders. These findings were supported by McDowell [185] who also noted that currently, many children with CVI are being supported using approaches and strategies that have been developed for children with ocular visual impairments despite current evidence indicating that many of these interventions may not be efficacious for children with CVI and may be detrimental for them, in some instances.

Visual field luminance thresholds were measured by Werth and Seelos [187] after children with CVI underwent systematic visual field training. They studied 17 children who had sustained blindness secondary to asphyxic-ischemic lesions and had done so more than a year before training. The training consisted of the stimulation of the blind visual field of these patients by slowly moving a light (velocity about 3 deg/s, luminance 26,000 cd/m^2^, diameter 5 mm) along the horizontal meridian and against a dark background from the periphery towards the center of the visual field. The training was given daily for about half an hour. After three months of the visual field training, visual functions developed in 11 of the children studied but no recovery was reported in any of the control participants (*N* = 37). The conclusion drawn by these investigators was that intact striatal and extra-striatal tissue can support visual function in childhood CVI. Little or negligible effects were noted in such training among adults [188].

Likova [189] provided support for the effect of early intervention on recovery of vision in children with early visual region insult. Likova reported on a case of V1 damage of a congenitally blind individual, who, before training, exhibited noisy, immature, and non-specific responses and after that training produced full-fledged response time-courses specific to the tactile-memory drawing task. The training consisted of an *Explore-Memorize* component, in which, employing only the left hand, the individual had to tactilely investigate a raised-line drawing model on the left part of a drawing tablet, and to develop from memory, a representation of the image to complete the *Memory-Draw* task. Likova then removed the model image. After 20 s, a stylus was employed by the participant to draw the image on the right part of the tablet with the right hand.

The results revealed the operation of a rapid training-based plasticity mechanism that recruits the resources of V1 in the process of learning to draw. The learning paradigm allowed Likova to investigate for the first time the evolution of plastic reassignment in V1 in a congenitally blind individual. These findings are consistent with a non-visual memory involvement of V1, and specifically, imply that the observed cortical reorganization can be empowered by the process of learning to draw.

While little if any interventional strategies have come from the clinical world, many novel suggestions have arisen from laboratory-based studies that may well gain traction in the clinic related to CVI in general and even seemingly to non-related developmental issues such as dyslexia. We have discussed how secondary visual pathways can mediate residual vision (blindsight) after V1 injury by relaying signals directly into higher-order cortical areas. Rima and Schmid [190] have suggested that dysfunction of secondary visual pathways might contribute to dyslexic’s perceptual difficulties and secondary visual pathways involving koniocellular LGN may be critical for the detection of visual change, whereas pulvinar function appears more linked to visuomotor planning.

## 8. Conclusions

The elaborate network oftentimes referred to as the brain’s “wiring” or “circuitry” is known to expand exponentially during fetal, neonatal and infant development. If connectivities are not maintained to repeatedly employed they are oftentimes eliminated and conversely when employed and firing together we can create novel functional connectivities, a process we understand as neuroplasticity. This function, explained in detail elsewhere [191,192,193] supports the notion that early childhood is the time to build either a strong and supportive, or fragile and unreliable foundation, which continues into childhood, adolescence, and adulthood.

The capacity for vision develops early in the development of the fetus and therefore in life. Reid and colleagues [194] confirmed that a fetus of 34-weeks GA can rotate the head to track patterned visual stimuli projected into the uterine environment. They concluded that visual-motor development is facilitated by a “gestational clock” rather than by, as most think, interactive visual experience that facilitates neuroplasticity in the visual system. There inevitably must exist a relationship between the fetus’s environment and visual development. During the third trimester of pregnancy, the vulnerability of white matter damage can be significant in which the optic radiations may be sensitive to insult. This is largely evidenced by unilateral periventricular hemorrhagic infarcts as well as bilateral ischemic lesions. It is during the third trimester of pregnancy that we note the significant degree of plasticity in the development of thalamocortical afferents, as it is that during this period, with relevant axonal guidance, that there still exists migration of afferents from the sub- to the cortical plate.

We had earlier discussed a case reported by Seghier and colleagues [40,41], of a three-month-old infant who had sustained a perinatal left arterial stroke with damage to the optic radiations but with sparing of the primary visual cortex PVC. Initial testing revealed cortical activation on the non-lesioned side with no evidence of the optic radiations on the ipsilesional side. When a follow-up study was performed at twenty months of age, *f*MRI demonstrated clear activation on the ipsilesional side representing neuroplastic changes in the thalamocortical pathway with attendant functional connectivities in the lateral geniculate. We can conclude from this and other cases that early insult to the PVC or optic pathways provides the infant with a critical period of developing neuroplastic changes that are better capable of allowing for the development of functional vision and with support for the conclusion that the child’s brain has a better capacity for rewiring than that of the older brain. The hypothesized mechanism for the relatively low prevalence of visual abnormalities post-stroke in children as compared to adults is assumed to be due to neuroplasticity within the visual system.

Congenital or neonatal brain damage invariably leads to bilateral injury commencing in the periolandic areas and involving gray matter and cortical regions of the brain but with the capacity for restoration of aspects of vision. In particular, we have learned that there oftentimes exists involvement of the geniculostriatal pathway, a specific loss of vision that is independent of any other functional abnormality, and that the regaining of the formerly impaired function with concomitant empirical change in brain state or electrophysiological activity is evidenced.

Besides understanding developmental aspects of visual neuroplasticity, we also examined the Sprague Effect [78] in which small tectal lesions can reestablish visual orientation in the half field contralateral to the lesion in animals with significant unilateral geniculostriatal impairment. More specifically, the Sprague Effect can be better understood by an understanding of its relationship to hemispatial neglect, a failure to react to novel stimuli presented to aspects of visual space, and without being able to attribute the lack of responsivity to a specific lesion.

The consequence of our discussion leads to an understanding that in those with central visual field defects, extrastriatal visual connectivities are greater when a lesion occurs earlier in life as opposed to when it occurs in the neurologically mature adult. The result is a significantly more optimized system of visual and spatial exploration within the ‘blind’ field of view. This then can serve as a basis for developing interventional schemes in congenital visual system insult.

## Figures and Tables

**Figure 1 brainsci-11-01279-f001:**
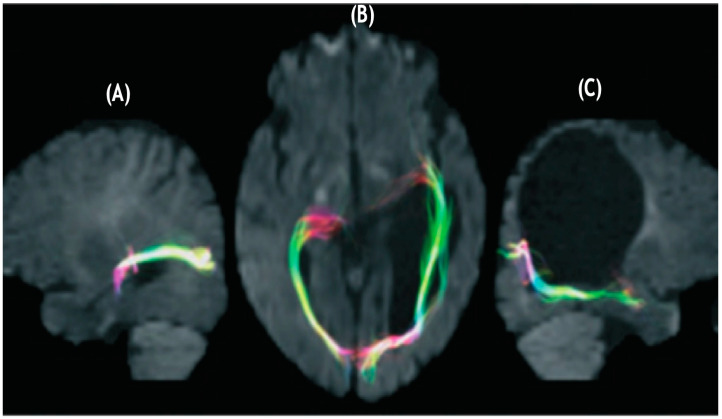
Diffusion tensor tractography of (**A**) optic radiations of left hemisphere evidencing a pathway avoiding an enlarged left ventricle projecting to the calcarine cortex. (**B**) bilateral optic radiations with the fibers of the left hemisphere proceeding laterally and anteriorly in contradistinction to the non-lesioned hemisphere. (**C**) Optic radiations of the right hemisphere appear to follow a normal trajectory (after Guzzetta et al. [56] with permission).

**Figure 2 brainsci-11-01279-f002:**
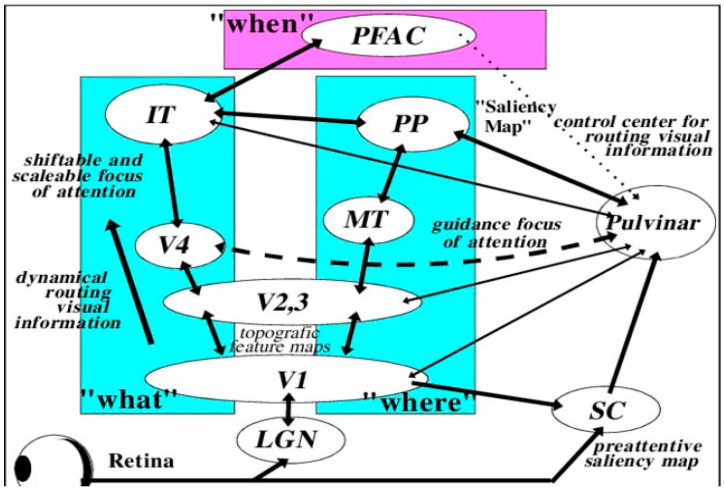
Major visual processing pathways of the primate brain considered in Gross et al.’s [99] model. Information from the retino-geniculostriatal pathway enters the visual cortex through area V1 and then proceeds through a hierarchy of visual areas that can be subdivided into two major functional pathways. The so-called “what”-pathway leads through V4 and the inferotemporal cortex (IT) and is mainly concerned with object-feature identification, regardless of position or size. V4 is the third area in the ventral stream obtaining strong feedforward signals from V2. Additionally, it receives projections directly from V1. The “where” pathway leads into the posterior parietal areas (PP) and is concerned with the locations and spatial relationships among objects, regardless of their identity. The “when” pathway involves the integration of signals from “What” and “Where” allowing for preplanning of movement and therefore response. (PFAC, prefrontal association cortex; IT, inferotemporal cortex; PP, posterior parietal areas; MT, middle temporal visual area; LGN, lateral geniculate nucleus; SC, superior colliculus) (after Gross et al. [99] with permission).

**Figure 3 brainsci-11-01279-f003:**
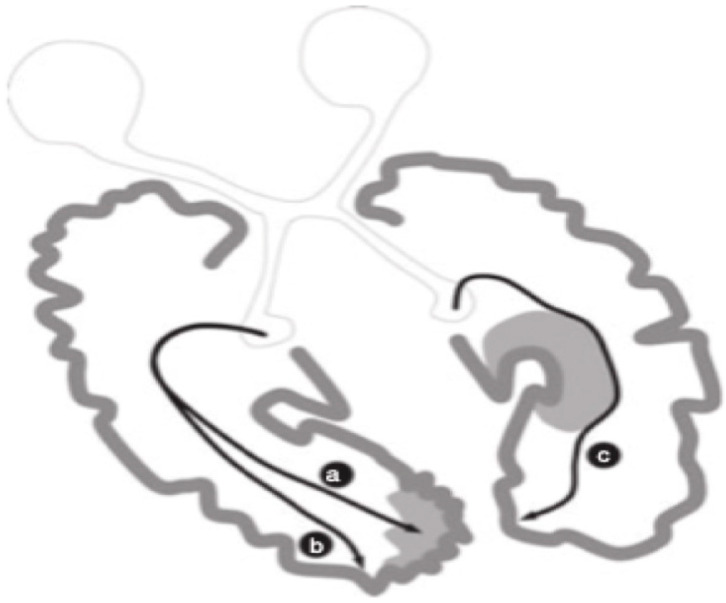
Potential mechanisms of neuroplasticity-based functional reorganization supporting normal visual function in congenitally brain-damaged individuals. (**a**) Represents damage to the PVC with functioning tissue existing within the lesion (**b**) aa reorganization occurring in regions external to the accepted boundaries of the PVC; (**c**) the geniculostriatal pathway bypassing the lesion and projecting to the calcarine cortex (after Guzzetta et al. [47] with permission).

**Figure 4 brainsci-11-01279-f004:**
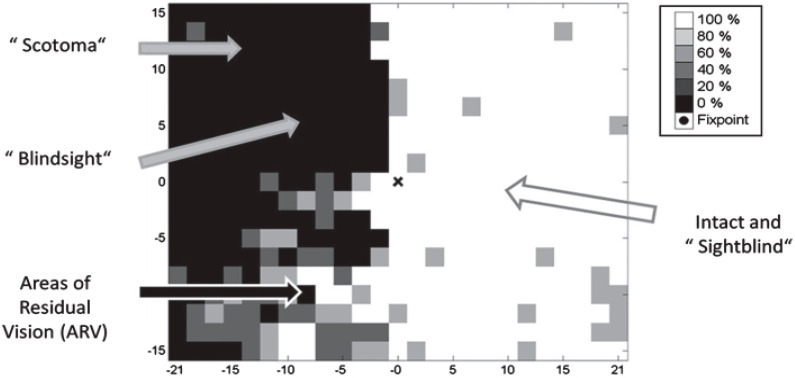
Subtle deficits and residual functions in visual fields with CVI. The phenomenon of blindsight occurs when otherwise blind individuals can correctly guess that stimuli were presented without the individual’s awareness of them. CVI individuals can correctly respond to visual stimuli, but they report seeing nothing (unconsciously seeing without knowing). These are areas in the visual field which are neither blind nor seeing normally (areas that are blind (black), partially impaired (grey), or normal (white)). Visual fields typically have different “shades of grey” where function is neither completely lost nor normal. Here blind individuals only occasionally respond to stimulation. With repeated testing, these visual field regions are variably responsive and are associated with increased thresholds and longer response time. X and Y axes represent the degree of angular subtense from central fixation. The concealed deficits in the “seeing field” effectively render individuals “sightblind”, measured by examinations that can measure higher cognitive dysfunctions. Especially clinically relevant is the grey “area of residual vision”, where vision is neither normal nor absent. These are the regions of the visual field with the greatest recovery potential. (from Sabel et al. [146], with permission).

**Figure 5 brainsci-11-01279-f005:**
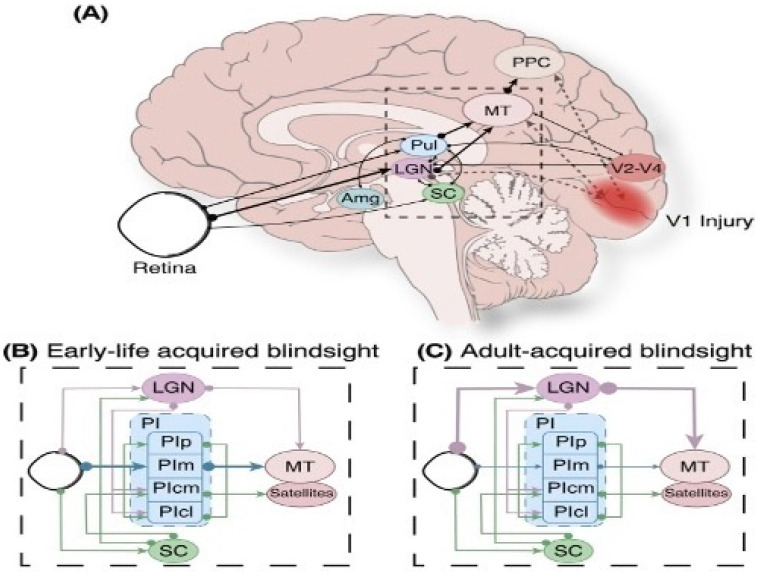
Subcortical pathways capable of facilitating blindsight. (**A**) Medial view demonstrating retinal extrastriatal region feedforward pathways in the posterior parietal cortex. Opaque pathways designate V1 injury neuronal degeneration. (**B**,**C**) Hypothesized subcortical pathways enabling blindsight after damage in V1 in the child’s (**B**) and in the adult brain (**C**). The strength of the projection is represented by line thickness. (Abbreviations: Amg, Amygdala; LGN, lateral geniculate nucleus; MT, middle temporal area; PI, inferior pulvinar; PIcl, caudolateral division of inferior pulvinar; PIcm, centromedial division of inferior pulvinar; PIm, medial division of inferior pulvinar; PIp, posterior division of inferior pulvinar; PPC, posterior parietal cortex; Pul, pulvinar; SC, superior colliculus) (from Fox at al., [149] with permission).

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
