# Peer review of "Cortical Visual Impairment in Childhood: ‘Blindsight’ and the Sprague Effect Revisited"

_brainsci, 2021, doi:10.3390/brainsci11101279_

Round 1
Reviewer 1 Report
In the paper “The Potential for the Rehabilitation of Cortical Visual Impairment in Childhood” the authors examined the influence of brain damage on brain plasticity and associated visual functions in humans. To this aim the authors reviewed the relevant literature about cortical visual impairment. The manuscript is well written and structured. The only minor remark is that the selected literature is understandingly biased towards relatively older papers, but still comprises only four papers from 2021. For instance, it could be beneficial to mention some relevant papers from 2021 describing recent methods (Itzhak et al., Child Neuropsychol), possible re-adaptations (Pamir et al., Neuropsychologia), and broad anatomo-functional networks (Ionta 2021, Front Hum Neur) associated with cerebral visual impairment. In this vein, the present paper could provide new perspectives on how to further ameliorate assessment and intervention procedures, fostering a better focus on possibly customized therapeutic approaches as a function of patients' characteristics.
Author Response
We thank the reviewer for the critical comments and obviously for the generally favorable review.
We have added a marked copy of the revised manuscript for the Reviewer's convenience
At the reviewer's suggestion, we have added relevant references that have appeared in the literature within the last twelve months. We deliberately did not want to provide structured intervention strategies as the journal is not a clinical publication and there exist numerous impediments to providing clinical guidance overtly - especially in the USA - which these authors have previously experienced. However, there are numerous interventional references in the paper that astute readers can avail themselves of - point taken though.
As a result of the Reviewer's suggestion, we have added a new section 6 entitled, "Possible Interventions in Childhood CVI". In that section, we have also taken the reviewer's suggestions and have added recent references relating to intervention.
The Reviewer's suggestion to include Pamir (2021) was not taken as is not a relevant paper inasmuch as this author discusses verbal mediation and not the issue of neuroplasticity directly and is also an N=1 of an adult and not a child.
Reviewer 2 Report
The work of Leisman and colleagues is a potentially really interesting review of literature about the role of rehabilitation on cortical visual impairment.
Unfortunately, I found this work really hard to read, maybe in part because I’m not a clinician and on the other hand because this work seems to be poorly written.
It contains a lot of words repetitions and typos and it is in general hard to read.
I can’t understand very well what is the real aim of the review.
The title and the body of the text are apparently in contradiction. The title states that the main topic is the rehabilitation of visual function, the abstract states that the main topic is brain plasticity. Please change the title of this review, it does not fit well with the content of the paper.
The organization of the text is chaotic.
The introduction paragraph is too general and does not introduce the reader to the work and to the aim of the study. The part on the development of child vision is more clearly written (relatively) but I can’t really understand why has been included and why it is so long.
I appreciated Chapter 3 even if 3.3 is really poorly written.
The rest of the review seems to be sufficiently interesting but I can’t clearly see a focus on rehabilitation.
There is a total lack of systematization or organization of literature in a meaningful way that helps the reader to be introduced to the topic.
Also, figures are problematic, captions are not sufficiently descriptive and not all the acronyms are defined. Certain figures require axis labeling.
For me, all the content should be massively reorganized and synthetized.
Other issues
258 The paragraph, albeit really interesting, is very difficult to read. It seems to contain ambiguous statements. Please rephrase. For example, the sentence in 265-270 for me is very difficult to understand. I don’t know a few of the techniques listed, what is “contrast sensitivity eye movements”? please add references to all the techniques mentioned.
240-242 for me this sentence is really hard to read and has no apparent meaning. Please rephrase and clarify
356-363 this sentence is organized badly, there is a parenthesis that never closes and I can’t figure why.
639 “Apparently” without sight? He is anophthalmic… I would remove the word apparently. He has visual concepts, but not vision.
658-661 This study is reported very badly. This review is supposed to be a review on rehabilitation, but here all the rehabilitation process is described with the word “training”.
What kind of training?
Figure 2
- Absence of references of what LGN, PFAC, SC acronym stand for
- The caption does not mention the “when” pathway
Figure 4
- no axes labels, are they degrees or what?
- heatmap has no label, what variable is quantified in gray levels?
Author Response
We thank the Reviewer for the thoughtful review and of course for the generally favorable comment about the overall work.
Please note that we are uploading a marked copy of the revised paper for the Reviewer's convenience.
Our responses to the comments are as follows:
- The work of Leisman and colleagues is a potentially really interesting review of literature about the role of rehabilitation on cortical visual impairment. Unfortunately, I found this work really hard to read, maybe in part because I’m not a clinician and on the other hand because this work seems to be poorly written. It contains a lot of words repetitions and typos and it is in general hard to read.
We thank the reviewer for the critical comments and have endeavored to make the article more readable, by initially having the “word repetitions and typos” expunged. We apologize for not having done so earlier.
The lead author is not a clinician either and as communication is 100 percent the responsibility of the sender, the onus is upon us to generate clarity. We trust that we have done so by a) addressing the typos etc., b) by adding a section on potential application and c) by revising the paper for better organizational clarity.
We have addressed the Reviewer’s examples of lack of clarity in seriatim. The overall paper’s revisions will be indicated later.
2. I can’t understand very well what is the real aim of the review. The title and the body of the text are apparently in contradiction. The title states that the main topic is the rehabilitation of visual function, the abstract states that the main topic is brain plasticity. Please change the title of this review, it does not fit well with the content of the paper.
The title of the paper has been changed so that it is hopefully concordant with the sum and substance of the review. The title is now, “Cortical Visual Impairment in Childhood: 'Blindsight’ and the Sprague Effect Revisited." The term “rehabilitation” has been removed from both the title and the abstract as the focus of the paper is more in keeping with neuroplasticity rather than active rehabilitative processes.
3. The introduction paragraph is too general and does not introduce the reader to the work and to the aim of the study.
The reviewer’s point is that the introductory comments are too general. We thought that the stage needs to be set for describing the nature of neurodevelopmental flexibility during fetal, neonatal, and infant development as the basis for neuroplastic changes in both cell connectivities (from sparse connections early in development to exuberant connectivities in childhood and paring down of the connectivities in adulthood (as illustrated below).
Elevated regional cerebral metabolic rate for glucose consumption (CMRGlc) during 3-10 yrs. corresponds to the era of exuberant connectivity needed for the energy needs of neuronal processes > by a factor of 2 compared to adults. PET shows relative glucose metabolic rate. We see the complexity of dendritic structures of cortical neurons consistent with the expansion of synaptic connectivities and increases in capillary density (in this case in the frontal cortex).
We have, however, added to the general introduction to make the purpose of the review more clear to the reader. In addition, the introductory section contains a very pointed statement indicating that "The development of the brain and the body consists of reciprocal feedback systems. Maturation of a particular sector of the brain stimulates activity in a corresponding area of the body or in a connected area of the brain. The stimulated function then matures more rapidly through use, but this use stimulates the development of the area of the brain that controls it." This, we contend is the justification for looking again at Sprague (1966) and of course everything following from Larry Weizkrantz and blindsight now in the context of Cortical Visual Impairment in childhood.
4. The part on the development of child vision is more clearly written (relatively) but I can’t really understand why has been included and why it is so long. I appreciated Chapter 3 even if 3.3 is really poorly written. The rest of the review seems to be sufficiently interesting but I can’t clearly see a focus on rehabilitation.
The reason for including the two subsections on normal visual development is, we think, that there needed to be a description of how the development of the visual system and its pathways develop and what the associated behavioral consequences of vision development in infancy are. This, we think, is necessary to juxtapose normal visual system development against alternative or collateral pathways that can develop after cerebral insult and how behavioral function could benefit from that.
All of this notwithstanding, we have shortened the section on vision development in neurotypical fetuses, neonates, and infants by approximately one-half.
Invariably situations arise in which multiple reviewers provide comments and suggestions that are diametrically opposed to each other. In this case, we have taken the middle position in which we have shortened the visual development section and made it more relevant to the general flow of the argument about developmental neuroplasticity but left the section (that we still think is unnecessary) on abnormal plasticity.
References to rehabilitation have been removed throughout the ms.
5. Also, figures are problematic, captions are not sufficiently descriptive and not all the acronyms are defined. Certain figures require axis labeling.
We thank the reviewer for pointing out deficiencies in labeling the graphic materials. Omissions and errors have been addressed.
6. Other issues
258 The paragraph, albeit really interesting, is very difficult to read. It seems to contain ambiguous statements. Please rephrase. For example, the sentence in 265-270 for me is very difficult to understand. I don’t know a few of the techniques listed, what is “contrast sensitivity eye movements”? please add references to all the techniques mentioned.
We also found it difficult to read and as result have rewritten the section no being reflected in lines 357-416.
240-242 for me this sentence is really hard to read and has no apparent meaning. Please rephrase and clarify.
We have removed the opening sentence and have left the section on negative aspects of neuroplasticity, per the request of another reviewer. (lines 333-357)
356-363 this sentence is organized badly, there is a parenthesis that never closes and I can’t figure why.
639 “Apparently” without sight? He is anophthalmic… I would remove the word apparently. He has visual concepts, but not vision.
Done! (line 878)
658-661 This study is reported very badly. This review is supposed to be a review on rehabilitation, but here all the rehabilitation process is described with the word “training”. What kind of training?
This section has been rewritten to include greater detail of the procedures used. (lines 910-922)
Figure 2
- Absence of references of what LGN, PFAC, SC acronym stand for
- The caption does not mention the “when” pathway
Now it does (cf. lines 429-508)
Figure 4
- no axes labels, are they degrees or what?
- heatmap has no label, what variable is quantified in gray levels?
These issues have been addressed in the text (lines 668-682)
Round 2
Reviewer 2 Report
For me, this review, albeit better than in the previous version, requires substantial rearrangements of structures, grammar, and syntax.
Please simplify sentences and don't repeat concepts over and over again.
The review should be shorter and straight to the point. Each paragraph you introduce should be instrumental to better understand the next section. Please remove (or fuse) and shorten all paragraphs non-direcly connected to the topic.
Here are some comments:
The introduction, albeit improved significantly from the original manuscript does not have any connection with the title. This is a review of Cortical visual impairment… Here I see only an introduction on brain development. I cannot figure why it is here. Here I would like to read why the authors wrote this review, what is the percentage of the diseases in the population, what would be the benefit to continuing the research in this field, and so on. In other words the background and rationale on why I need to read this work.
1.1 This chapter is really interesting but also here I can’t see how it is connected to the main topic, It can be eliminated or greatly reduced. I think that only a paragraph can be spent on visual development, the rest has no reason to exist. It gets really long and has nothing to do with the meaning of why the reader is reading this work.
1.2 Same as 1.1 The paragraph is interesting but for me is too long and the reader gets bored of so much information without an apparent reason.
1.3 Same as above
2
In the paper, there are no references to the concept of plasticity. What is plasticity? Please add one sentence or two in the introduction.
3.1 The initial part of the paragraph is very hard to read, it needs a general rearrangement and simplification of sentences. I hardly can figure what are you talking about.
3.2 Albeit of interest, this paragraph is useless. Why do I need to read this in this review? I cannot understand why you spent an entire paragraph talking about “negative” plasticity on alcohol consumption in a review on visual impairment. Remove this please or fuse (one sentence or two) with 3.1
3.3 Why this is a sub-paragraph? I think this should be an independent part and not a sub-paragraph of the neuroplasticity part. This is the main topic of the review...
249 What is the like? I cannot figure what you mean by "contrast sensitivity eye movements", please explain. Reference 73 refers to methods for early child visual assessment. Maybe you mean contrast sensitivity assessed using eye movements?
Author Response
The paper has been substantively revised - again.
The responses are provided in seriatim.
For me, this review, albeit better than in the previous version, requires substantial rearrangements of structures, grammar, and syntax.
This has been done and a marked copy has been provided for convenience
Please simplify sentences and don't repeat concepts over and over again.
Again, the paper has been thoroughly revised and reviewed and substantive changes made - especially with a view towards clarity and the linear flow of ideas.
The review should be shorter and straight to the point. Each paragraph you introduce should be instrumental to better understand the next section. Please remove (or fuse) and shorten all paragraphs non-directly connected to the topic.
About 25 percent of the ms. has been chopped especially the section on the development of normal fetal, neonatal and infant vison
Here are some comments:
The introduction, albeit improved significantly from the original manuscript does not have any connection with the title. This is a review of Cortical visual impairment… Here I see only an introduction on brain development. I cannot figure why it is here. Here I would like to read why the authors wrote this review, what is the percentage of the diseases in the population, what would be the benefit to continuing the research in this field, and so on. In other words the background and rationale on why I need to read this work.
The reason for the inclusion of normal division development from fetus onward was largely a function of other reviewers (of which there were four) we agree with the current reviewer's suggestion and have removed the material that was suggested to be excluded.
Epidemiological references and content has been included in the introductory section with the indication that CVI is a significant global health issue. The justification for producing the reports is now given clearly
1.1 This chapter is really interesting but also here I can’t see how it is connected to the main topic, It can be eliminated or greatly reduced. I think that only a paragraph can be spent on visual development, the rest has no reason to exist. It gets really long and has nothing to do with the meaning of why the reader is reading this work.
The vision development material has been removed and is headed for our reports of our work in fetal cognition and sensation. Not here!
1.2 Same as 1.1 The paragraph is interesting but for me is too long and the reader gets bored of so much information without an apparent reason.
Removed
1.3 Same as above
Removed!
2
In the paper, there are no references to the concept of plasticity. What is plasticity? Please add one sentence or two in the introduction.
This section has been greatly expanded!
3.1 The initial part of the paragraph is very hard to read, it needs a general rearrangement and simplification of sentences. I hardly can figure what are you talking about.
Revised!
3.2 Albeit of interest, this paragraph is useless. Why do I need to read this in this review? I cannot understand why you spent an entire paragraph talking about “negative” plasticity on alcohol consumption in a review on visual impairment. Remove this please or fuse (one sentence or two) with 3.1
Negative plasticity was added to the paper at the request of another reviewer. We agree with this comment and as a result, have removed the section but included a few sentences under the general heading of neuroplasticity.
3.3 Why this is a sub-paragraph? I think this should be an independent part and not a sub-paragraph of the neuroplasticity part. This is the main topic of the review...
Changed!
249 What is the like? I cannot figure what you mean by "contrast sensitivity eye movements", please explain. Reference 73 refers to methods for early child visual assessment. Maybe you mean contrast sensitivity assessed using eye movements?
Correct. The error has been fixed!